# *In Vitro* and *In Vivo* Effect of the Imidazole Luliconazole against *Lomentospora prolificans* and *Scedosporium* spp.

Dan-Tiberiu Furnica,[a] Silke Dittmer,[a] Ulrike Scharmann,[a] Jacques F. Meis,[b,c] Joerg Steinmann,[a,d] Peter-Michael Rath,[a] Lisa Kirchhoff[a]

[a]Institute of Medical Microbiology, Excellence Center for Medical Mycology (ECMM), University Hospital Essen, University of Duisburg-Essen, Essen, Germany
[b]Department of Internal Medicine, University of Cologne, Faculty of Medicine and University Hospital Cologne, Excellence Center for Medical Mycology (ECMM), Cologne, Germany
[c]Excellence Center for Medical Mycology (ECMM), Centre of Expertise in Mycology, Radboudumc/Canisius Wilhelmina Hospital, Nijmegen, The Netherlands
[d]Institute of Clinical Hygiene, Medical Microbiology and Infectiology, Klinikum Nürnberg, Paracelsus Medical University, Nuremberg, Germany

**ABSTRACT** Infections with *Scedosporium* spp. and *Lomentospora prolificans* have become a serious threat in clinical settings. The high mortality rates associated with these infections can be correlated with their multidrug resistance. The development of alternative treatment strategies has become crucial. Here, we investigate the *in vitro* and *in vivo* activity of luliconazole (LLCZ) against *Scedosporium apiospermum* (including its teleomorph *Pseudallescheria boydii*) and *Lomentospora prolificans*. The LLCZ MICs were determined for a total of 37 isolates (31 *L. prolificans* isolates, 6 *Scedosporium apiospermum*/*P. boydii* strains) according to EUCAST. Furthermore, the LLCZ antifungal activity was tested *in vitro*, using an XTT [2,3-bis-(2-methoxy-4-nitro-5-sulfophenyl)-2H-tetrazolium-5-carboxanilide salt] growth kinetics assay and biofilm assays (crystal violet and XTT assay). In addition, a *Galleria mellonella* infection model was used for *in vivo* treatment assays. The MIC$_{90}$ of LLCZ was determined to be 0.25 mg/L for all tested pathogens. Growth was inhibited within 6 to 48 h of the start of incubation. LLCZ inhibited biofilm formation in both preadhesion stages and late-stage adhesion. *In vivo*, a single dose of LLCZ increased the survival rate of the larvae by 40% and 20% for *L. prolificans* and *Scedosporium* spp., respectively. This is the first study demonstrating LLCZ activity against *Lomentospora prolificans in vitro* and *in vivo* and the first study showing the antibiofilm effect of LLCZ in *Scedosporium* spp.

**IMPORTANCE** *Lomentospora prolificans* and *S. apiospermum*/*P. boydii* are opportunistic, multidrug-resistant pathogens causing invasive infections in immunosuppressed patients and sometimes in healthy persons. *Lomentospora prolificans* is panresistant against the currently available antifungals, and both species are associated with high mortality rates. Thus, the discovery of novel antifungal drugs exhibiting an effect against these resistant fungi is crucial. Our study shows the effect of luliconazole (LLCZ) against *L. prolificans* and *Scedosporium* spp. *in vitro*, as well as in an *in vivo* infection model. These data reveal the previously unknown inhibitory effect of LLCZ against *L. prolificans* and its antibiofilm effect in *Scedosporium* spp. It represents an extension of the literature regarding azole-resistant fungi and could potentially lead to the development of future treatment strategies against these opportunistic fungal pathogens.

**KEYWORDS** biofilm, *Lomentospora*, *Scedosporium*, azole resistance, luliconazole, antifungal therapy

Address correspondence to Dan-Tiberiu Furnica, dan-tiberiu.furnica@uk-essen.de.

The authors declare no conflict of interest.

In the last decades, infections with fungal pathogens have become more prevalent in clinical settings, especially in vulnerable patient groups (1). Some clinically relevant fungal pathogens possess an increased resistance to commonly administered antifungal

drugs (2), with incidence rates of these multidrug-resistant pathogens increasing at a concerning pace (3). *Lomentospora prolificans* is a fungal pathogen known for its resistance against multiple anti-infective agents belonging to different substance classes, such as echinocandins, pyrimidines, allylamines, polyenes, and azoles (to a limited extent) (4). *Scedosporium apiospermum* and *Pseudallescheria boydii* are known for their increased resistance to amphotericin B (2, 5). For these organisms, conventional treatment with triazoles showed only a reduced response rate from the patients enlisted in clinical trials (6). In the case of *L. prolificans*, there is currently no standardized treatment available. The current preferred treatment option is voriconazole (VCZ), mostly in combination therapy, as suggested on the basis of both *in vitro* data and clinical studies (7, 8). However, these therapies yield variable results (9–11). The mortality rates of patients with *L. prolificans* disseminated infections and underlying hematological/oncological malignancies remain high in most clinical studies (12, 13). Among other factors, the biofilm formation capability of fungi is linked to their increased resistance against conventional treatments (14). Biofilms are known for their ability to offer protection against the host's immune system, resistance to antimicrobial agents, or increased virulence (15). Like other filamentous fungi such as *Aspergillus fumigatus*, *L. prolificans* and *Scedosporium* spp. are known to adhere as conidial cell forms onto both living and inert substrata, followed by germination of the conidial cells and differentiation into a three-dimensional biofilm-like structure (16). For this reason, the development of viable strategies for the treatment of *L. prolificans* and *Scedosporium* spp. has become imperative. As part of a separate project, 400 compounds were donated by the Medicines for Malaria Venture in the form of a Pandemic Response Box. The potential antimicrobial activities of these compounds were screened against several fungi, including *L. prolificans* and *Scedosporium* spp.

One compound that has shown activity against *L. prolificans* and *Scedosporium* spp. is luliconazole (LLCZ). Clinically, LLCZ is well known for its effect against dermatophytes (e.g., *Trichophyton* spp.) and is currently approved only for topical use in the treatment of superficial mycoses such as tinea pedis, tinea cruris, tinea corporis, and onychomycosis (17–19). *In vivo*, its effect against filamentous fungi (such as *Aspergillus niger* or *A. fumigatus*, including azole-resistant strains) has also been documented (20–23). Some recent publications have also demonstrated the effect of LLCZ against the planktonic growth of *Scedosporium* spp. (24, 25). In the case of *A. fumigatus*, the effect of LLCZ against biofilm formation has also been demonstrated (especially against early stage biofilm) (23). Similar to triazoles, LLCZ affects ergosterol synthesis by inhibition of lanosterol demethylase. The *R*-enantiomer of this compound inhibits lanosterol demethylase activity, while the *S*-enantiomer shows no significant effect (26). However, it is yet unclear how its mechanism of action differs from that of triazoles.

In 2015, the Food and Drug Administration (FDA) approved LLCZ for topical use in the United States (27). Further clinical studies are necessary to assess the safety profile of LLCZ for oral or intravenous (i.v.) use. Nonclinical studies show that the lethal dose for LLCZ is 2,000 mg/kg in rats, which would indicate a potentially safe pharmacokinetic profile of the drug (28). Aside from the unknown mechanism of action, and the unavailability of an oral formulation, LLCZ has also shown a poor solubility and retention in the skin. However, the drug shows no concerning side effects, and oral formulations have yielded good results in murine models during its development phase (29).

The aim of this study was to investigate the effect of LLCZ against the planktonic growth and biofilm formation of *L. prolificans* and *S. apiospermum*/*P. boydii in vitro* and *in vivo*.

## RESULTS

The overall MIC values ranged between 0.004 and >0.25 mg/L. Whereas the MICs for *L. prolificans* ranged between 0.004 and >0.25 mg/L, for *S. apiospermum*/*P. boydii*, the obtained values were between 0.008 and 0.06 mg/L. The overall $MIC_{90}$ and $MIC_{50}$ values were 0.25 mg/L and 0.06 mg/L, with no significant differences between the

**TABLE 1** List of the isolates included in this study and the isolate-specific MICs of LLCZ[a]

| Isolate ID | LLCZ MIC (mg/L) | Species | Source |
|---|---|---|---|
| **2053** | 0.25 | *L. prolificans* | CF[b] |
| **2054** | 0.06 | *L. prolificans* | CF[b] |
| **2055** | 0.03 | *L. prolificans* | CF[b] |
| **2058** | 0.25 | *L. prolificans* | CF[b] |
| **2449** | 0.06 | *L. prolificans* | CF |
| **2229** | 0.03 | *L. prolificans* | CF |
| **2340** | 0.06 | *L. prolificans* | BMT |
| F141 (CBS 100390) | 0.125 | *L. prolificans* | Lymphatic leukemia |
| 2049 | 0.03 | *L. prolificans* | CF[b] |
| 2050 | 0.008 | *L. prolificans* | CF[b] |
| 2051 | 0.03 | *L. prolificans* | CF[b] |
| 2052 | 0.06 | *L. prolificans* | CF[b] |
| 2056 | 0.25 | *L. prolificans* | CF[b] |
| 2057 | 0.125 | *L. prolificans* | CF[b] |
| 2059 | 0.25 | *L. prolificans* | CF[b] |
| 2060 | 0.004 | *L. prolificans* | CF[b] |
| 2061 | 0.06 | *L. prolificans* | CF[b] |
| 2062 | 0.06 | *L. prolificans* | CF[b] |
| 2063 | 0.125 | *L. prolificans* | CF[b] |
| 2064 | 0.015 | *L. prolificans* | CF[b] |
| 2065 | >0.25 | *L. prolificans* | CF[b] |
| 2066 | 0.125 | *L. prolificans* | CF[b] |
| 2067 | 0.015 | *L. prolificans* | CF[b] |
| 2068 | >0.25 | *L. prolificans* | CF[b] |
| 2069 | 0.03 | *L. prolificans* | CF[b] |
| 2070 | 0.015 | *L. prolificans* | CF[b] |
| 2071 | 0.125 | *L. prolificans* | CF[b] |
| 2073 | 0.25 | *L. prolificans* | CF[b] |
| 2074 | 0.25 | *L. prolificans* | CF[b] |
| 2075 | 0.25 | *L. prolificans* | CF[b] |
| 2076 | 0.06 | *L. prolificans* | CF[b] |
| **2381** | 0.03 | *S. apiospermum* | CF |
| **2387** | 0.008 | *S. apiospermum* | CF |
| **M222** | 0.03 | *S. apiospermum* | NA |
| **M356** | 0.015 | *P. boydii* | NA |
| **2391** | 0.008 | *P. boydii* | BAL |
| **F125** | 0.06 | *P. boydii* | CF |

[a]Isolates indicated in bold were used for the growth kinetics and biofilm assays. BMT, bone marrow transplant; CF, cystic fibrosis patient; BAL, bronchoalveolar lavage specimen; NA, not available.
[b]Obtained from the Dutch CF Fungal Collection Consortium (30).

distinct organisms (Table 1 and Fig. 1). A geometric mean MIC of 0.122 mg/L was calculated.

The activity of LLCZ against the planktonic growth of *L. prolificans* and *S. apiospermum*/*P. boydii* over time was analyzed with an XTT [2,3-bis-(2-methoxy-4-nitro-5-sulfophenyl)-2H-tetrazolium-5-carboxanilide salt] growth kinetics assay, revealing that various concentrations of the antifungal agent had a significant effect (Fig. 2). The inhibition started after 6 h, and a decrease in metabolic activity could be observed until 48 h, which was also the last measured time point.

The effect of LLCZ against biofilm was also analyzed via XTT and crystal violet (CV) assays at different formation stages. While the effect of the drug was similar for all three organisms, the results of the two assays differed. The XTT assay (Fig. 3) showed a decrease in metabolic activity of about 75% when the drug was added in the early stages of biofilm formation (0 h and 2 h), whereas the CV assay (see Fig. S1 in the supplemental material) showed a decrease in biofilm mass of 22% (0 h) and 50% (2 h). Similar results were recorded for late-stage biofilm. In both the XTT and CV assays, a biofilm inhibition of about 30% was demonstrated.

The effect of the antifungal agent could be visualized using confocal laser scanning microscopy (CLSM). *L. prolificans* isolate 2229 was treated with an LLCZ concentration of 0.03 mg/L (Fig. 4B) and half of this concentration (Fig. 4C). In both cases, the biofilm

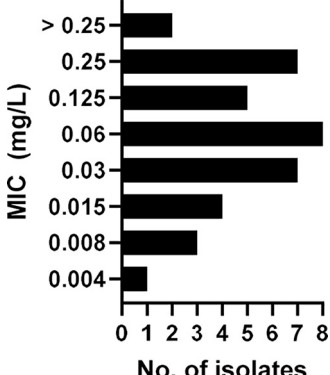

**FIG 1** MICs of LLCZ for the 31 *L. prolificans* isolates and 6 *S. apiospermum/P. boydii* isolates.

formation of *L. prolificans* was inhibited. Figure 4A shows the biofilm formation of the untreated *L. prolificans* isolate. Hyphal development is visible in Fig. 4C, where a sub-MIC was used, while the formation of hyphae was inhibited when treated with LLCZ at the MIC.

It was determined previously that LLCZ has no toxic effect on *Galleria mellonella* (23). An infection assay (Fig. S2) showed that the optimal fungal inoculum for all strains was $2 \times 10^7$ CFU/mL, with a survival outcome of 21% for *L. prolificans*, 10% for *S. apiospermum*, and 25% for *P. boydii* after a total incubation period of 14 days at 37°C. The larvae started to die 1 to 2 days after infection. When treated with a dose of 152 mg/L LLCZ, the survival outcome of the larvae increased significantly, compared to that of the placebo group. The treated larvae infected with *L. prolificans* showed a 40% higher survival outcome and those infected with *S. apiospermum* and *P. boydii* around 15% and 20% higher, respectively. VCZ was used as a negative control for *L. prolificans* and as a positive control for *S. apiospermum* and *P. boydii*. While treatment with VCZ

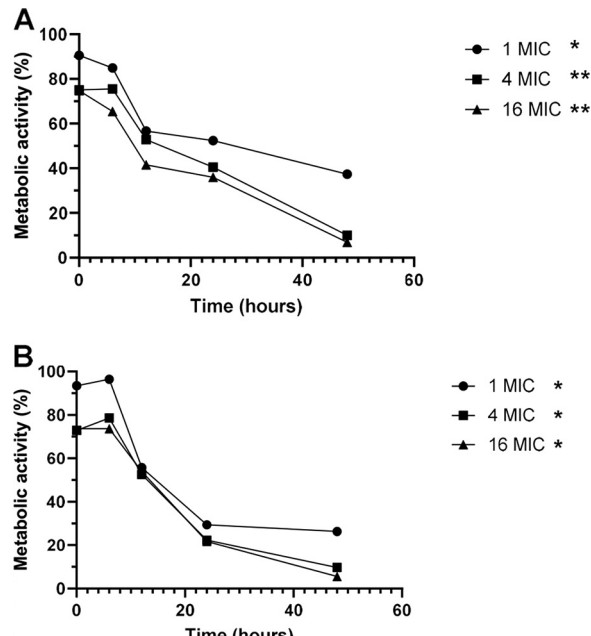

**FIG 2** Representation of the growth kinetics of the eight *L. prolificans* isolates (A) and six *S. apiospermum/P. boydii* isolates (B) treated with different concentrations of LLCZ. The organisms were incubated at 35°C for 48 h. The optical density at 492 nm ($OD_{492}$) was measured after 2 h, 6 h, 12 h, 24 h, and 48 h of incubation. Through incubation with 50 $\mu$L XTT (2.5 mg/mL) plus menadione (125 mg/mL) per well (the XTT was added 2 h prior to the incubation end), the metabolic activity of the cells was determined. Statistical significance was determined using Dunnett's multiple-comparison test. Significance was set at $P < 0.05$; *, $P < 0.05$; **, $P < 0.01$.

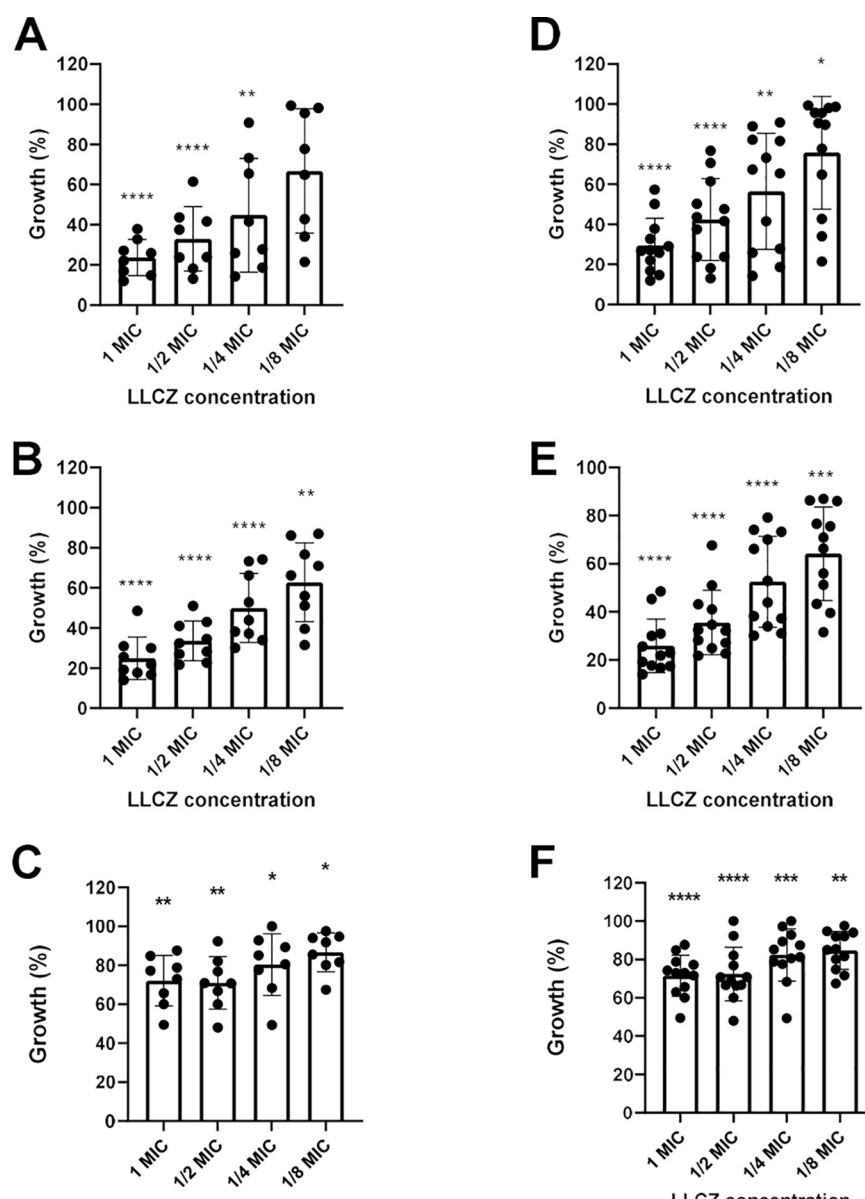

**FIG 3** (A to C) Growth of *L. prolificans* biofilm (8 isolates); (D to F) growth of all organisms (8 *L. prolificans* isolates and 6 *S. apiospermum/P. boydii* isolates) when treated with LLCZ (XTT assay). Isolates were treated with 1×, 0.5×, 0.25×, and 0.125× the isolate-specific MIC at 0 h, 2 h, and 48 h after incubation.

yielded survival increases of 15% in *S. apiospermum* and 25% in *P. boydii*, no significant effect of this antifungal agent could be seen in *L. prolificans* (Fig. 5).

## DISCUSSION

This study is the first to report the *in vitro* effect of LLCZ against both planktonic growth and biofilm formation of the azole-resistant mold *L. prolificans*. Furthermore, the effects of this antifungal agent were assessed in an *in vivo* invertebrate treatment model.

In the study, 37 isolates were tested according to the EUCAST protocol, showing a wide MIC range of between 0.004 and >0.25 mg/L. No significant differences or distinct patterns were observable in the MIC distribution of the three organisms. Although no comparable data exist for *L. prolificans*, LLCZ is known to have relatively low MIC values for *Scedosporium* spp.; other molds, such as *A. fumigatus* and non-*fumigatus* *Aspergillus* spp.; and also for yeasts, such as *Candida albicans* (21, 23–25, 31).

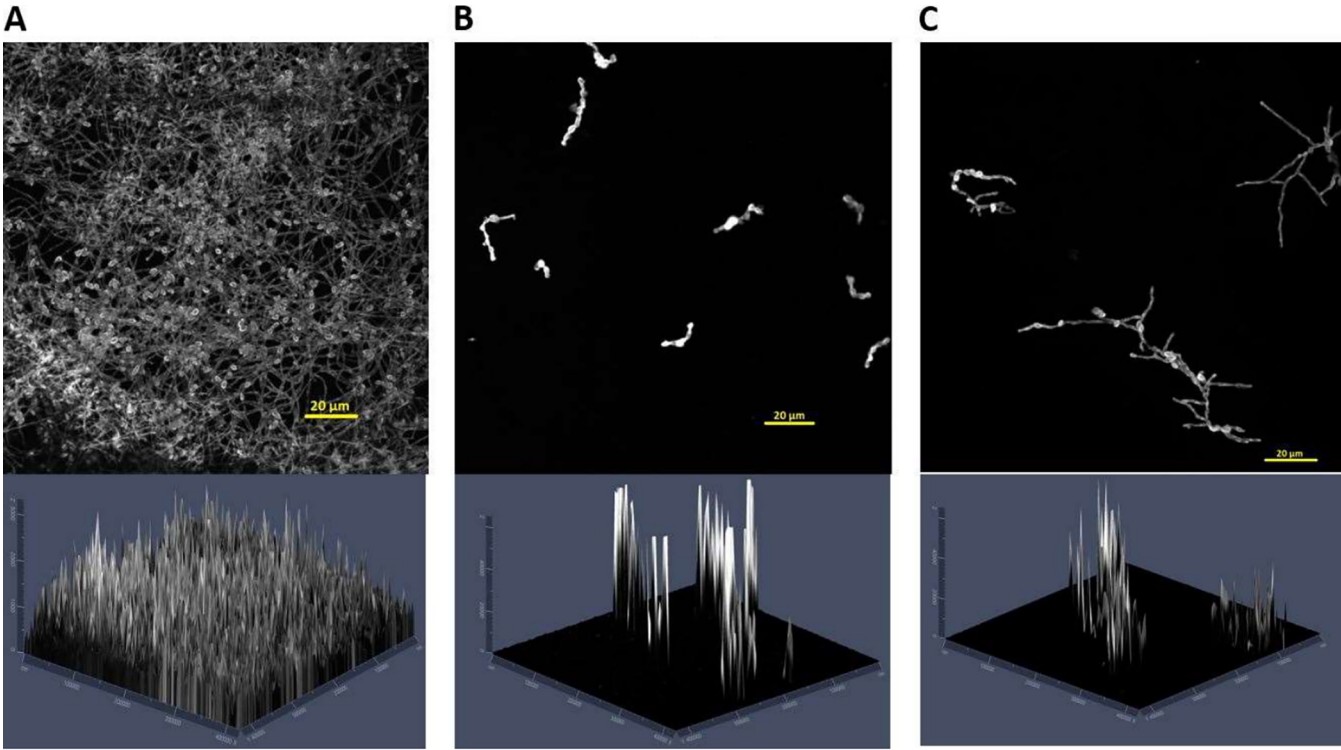

**FIG 4** CLSM 2D (top) and 2.5D (bottom) images of *L. prolificans* (isolate 2229) biofilm. Untreated (A); treated with the isolate-specific MIC (0.03 mg/L) 0 h after incubation start (B); treated with one-half the isolate-specific MIC (0.015 mg/L) 0 h after incubation start (C).

An XTT growth kinetics assay was used to determine the effect of different LLCZ concentrations against the planktonic growth of the organisms (Fig. 2). The metabolic activity of the LLCZ-treated cells was measured and compared to that of the untreated cells. The results showed a steady inhibition of planktonic growth between 6 h and 48 h. A stronger effect was observable at higher LLCZ concentrations. However, the MIC did significantly inhibit the metabolic growth of all organisms, showing the concentration-dependent activity of LLCZ. In contrast to the results from growth inhibition experiments of *A. fumigatus* with LLCZ, the growth inhibition of *L. prolificans* and *S. apiospermum*/*P. boydii* appeared to be continuous, and no evident activity recovery was recorded (23). This regenerative effect was also observable with *A. fumigatus* when applying other antifungal drugs, such as amphotericin B, VCZ, or olorofim (32). Although the growth inhibition for *L. prolificans*, *S. apiospermum*, and *P. boydii* seemed to be constant, the antifungal agent did not manage to reach an inhibition of >99%, showing that LLCZ has a fungistatic effect against these organisms (33). This is congruent with LLCZ activity against *A. fumigatus* (23).

The biofilm formation capacity of LLCZ-treated fungi was assessed via two separate assays. While the XTT assay showed a significant decrease in metabolic activity (75%) in all organisms when the drug was directly incubated with the fungal suspension (0 h), this could not be correlated with a decrease in biomass, since the CV revealed a decrease of only 22% of the total generated biofilm mass (see Fig. S1 in the supplemental material). When the drug was added at a later time point (2 h after incubation), the amount of produced biomass in biofilm was reduced by approximately 20%. This decrease could be attributed to the washing step prior to the addition of the drug (see "*In vitro* assays. [iii] Biofilm assays" in Materials and Methods, below). No difference was recorded in metabolic activity when the drug was added at 0 h or 2 h after incubation. It is possible that LLCZ effectively inhibits the metabolic activity and development of *L. prolificans* and *Scedosporium* spp., despite their high biofilm production. This may be the explanation for the significant effect of LLCZ treatment against mature biofilms (48 h after incubation). Comparable to LLCZ, the novel antifungal drug olorofim also showed a partial (40%) inhibition of *L. prolificans* and *Scedosporium* spp., but it has

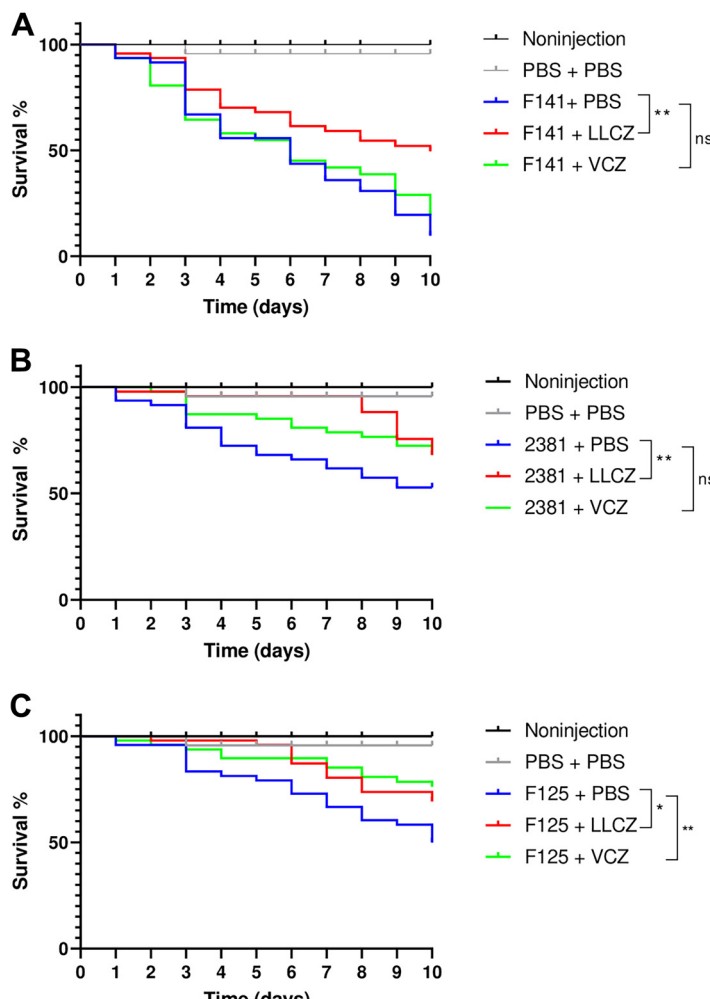

**FIG 5** Survival curves for *Galleria mellonella* larvae infected with *L. prolificans* (A), *S. apiospermum* (B), and *P. boydii* (C). The larvae were treated with PBS as a placebo (blue) and LLCZ (red) during a 10-day period. Voriconazole (VCZ) was used as a control for all organisms (green). Groups of 15 larvae were injected with a fungal inoculum of $2 \times 10^7$ CFU/mL and then after 48 h with 152 mg/L LLCZ (treatment), 160 mg/L VCZ (control), or PBS (placebo). The control groups were injected with PBS (gray) and then reinjected after 48 h, in order to rule out any false-positive results caused by the piercing damage of the needle or by LLCZ. A noninjection control (black) was used as an indicator of the quality of the larvae. The larvae were incubated at 37°C, and mortality was checked every 24 h for at least 10 days. Statistical significance of the survival curves was determined using a log-rank (Mantel-Cox) test. *, $P < 0.05$; **, $P < 0.01$; ns, not significant.

been described as ineffective against late-stage biofilm of azole-resistant *A. fumigatus* isolates (32, 34). CLSM imaging confirmed LLCZ activity against biofilm formation in *L. prolificans* when applied in the early formation stages, even though hyphae could still be detected, most notably when sub-MICs (half the MIC) were used. Due to the nature of the CV assay (staining biomass without distinction between dead or living cells), dead hyphae could be detected. However, according to the XTT assay, these hyphae were metabolically inactive.

In addition to the *in vitro* effect, this study also investigated the effect of LLCZ *in vivo* in the invertebrate infection model using *G. mellonella*. Despite literature recommendations ($1 \times 10^6$ CFU/mL), an inoculum of $2 \times 10^7$ CFU/mL was chosen for all three organisms, due to their rather reduced virulence (35). Treatment with a single dose of LLCZ 48 h after the initial infection showed a 40% increase in the survival of the larvae infected with *L. prolificans* and a 20% and 15% survival increase for the larvae infected with *P. boydii* and *S. apiospermum*, respectively, after 10 days. The therapeutic dose of 152 mg/L was adapted from a study in which rats infected with *A. fumigatus* were

treated with a daily dose of LLCZ (36). To the best of our knowledge, there are no previous clinical or nonclinical accounts of this drug being used to treat *L. prolificans* or *Scedosporium* spp. *in vivo*. Therefore, it is not known if this is an optimal therapeutic dose for these organisms. For VCZ, the recommended therapeutic dose is 6 mg/kg (i.v.) twice a day for two doses, followed by 4 mg/kg twice a day (37). Due to the nature of our larval model, multiple-dose treatments resulted in an increased mortality of the larvae through the repeated injections. This could have had a masking effect on the data and led to irreproducibility of the experiments. Despite the several advantages that *G. mellonella* offers, including the possibility of extensively testing a drug against multiple pathogenic organisms, its main drawback is the fact that the invertebrate larvae are physiologically very different from humans (38). Despite this, it is a reliable *in vivo* model, surpassing *Caenorhabditis elegans*, where direct injection is impossible, and it could be used prior to *in vivo* testing in vertebrate animal models (39).

Despite the overwhelmingly positive results in *in vitro* experiments, LLCZ is not yet approved by the FDA for systemic or oral use. In its current topical formulation, LLCZ is effective only against superficial skin mycoses (17). Due to the low solubility of the drug, treatment of a deep-seated infection is challenging because of its retention in the skin (29). This could mean that the development of a systemic LLCZ treatment will not be straightforward. Research is ongoing for the development of novel strategies to improve the drug's penetration and absorption capabilities. These strategies include the use of nanostructured lipid carriers, which show promising results (40).

Oral formulations of the drug have already been successfully tested in an *ex vivo* murine model and represent a potential solution to this problem (36). Alternatively, respiratory infections could be treated via aerosol inhalation of LLCZ. This approach has already been successfully used for other topical antifungal drugs such as *N*-chlorotaurine (41, 42). LLCZ showed no significant side effects in clinical and nonclinical trials; thus, the future testing and development of an aerosol or oral formulation could represent a potential treatment for *L. prolificans* and *Scedosporium* sp. infections (17, 28, 29).

As a limitation of this study, it must be noted that the number of *S. apiospermum* ($n = 3$)/*P. boydii* ($n = 3$) isolates was relatively low due to a lack of availability. Similarly, for the *in vivo* experiments, the larvae were infected with only one isolate of each organism. Further, the invertebrate animal model may also have been impacted by several external factors, such as light sensitivity and weather conditions, which may have impacted the viability of certain larval batches and influenced the reproducibility of the experiment (43). The use of a murine animal model to confirm the data obtained here could solve this issue. A further limitation is the fact that the genetic diversity of the isolates is unknown. It would be preferable for further experiments to include *L. prolificans* and *Scedosporium* sp. isolates with a confirmed genetic diversity. In conclusion, for future experiments, it is suggested to increase the total number of isolates, to include more strains/organisms which are confirmed to be genetically diverse, and to use a murine infection model, in order to properly validate the promising first results.

This study is the first account of the antifungal effect of LLCZ against planktonic growth of *L. prolificans*. Furthermore, it is the first study to show the effect of LLCZ against biofilm formation of *S. apiospermum*/*P. boydii* and to offer data for an *in vivo* treatment model of these fungal pathogens.

## MATERIALS AND METHODS

**Isolates.** The study comprised 31 *L. prolificans* isolates and 6 *S. apiospermum*/*P. boydii* strains (Table 1). These organisms are well known for their panresistance to antifungal drugs. Of the 31 *L. prolificans* isolates, 8 isolates (shown in bold in Table 1) were used for growth kinetics and biofilm assays. The organisms indicated were obtained from the Dutch CF Fungal Collection Consortium (30). No clinical information regarding the isolates is available.

**Fungal suspensions.** The fungal suspensions were prepared as described elsewhere (34). In brief, the isolates were plated on Sabouraud agar (Oxoid, Wesel, Germany) and incubated for 5 to 7 days at 30°C. In order to collect viable conidia, a solution of $H_2O$ plus 0.1% Tween was pipetted onto the plates. The fungal suspension was collected with a syringe and filtered to remove hyphae (pore size, 10 $\mu$m; filcon syringe; Becton Dickinson, Franklin Lakes, NJ, USA). After washing once with $1\times$ phosphate-buffered saline (PBS), the inoculum was adjusted to an appropriate concentration in RPMI 1640 medium.

**Luliconazole preparation.** LLCZ was purchased from Sigma-Aldrich (St. Louis, MO, USA). Dimethyl sulfoxide (DMSO) was used for dilution of the drug into stock solutions, which were stored at $-20°C$ until further use. The stock was diluted to various working concentrations, followed by a 1:100 dilution in RPMI 1640 medium. A DMSO concentration of 1% remained in the aliquots. For further information, refer to the corresponding sections below.

***In vitro* assays. (i) Broth microdilution.** The isolate-specific MIC of LLCZ was determined via broth microdilution according to the European Committee on Antimicrobial Susceptibility Testing (EUCAST) (44). LLCZ was diluted in DMSO and double-concentrated RPMI 1640 medium (2% glucose) to final concentrations ranging between 0.0004 and 0.25 mg/L. After an incubation time of 48 h, the MIC of every individual isolate was read visually.

**(ii) Growth kinetics.** To assess the effect of LLCZ against the planktonic growth of *L. prolificans* (8 isolates) and *S. apiospermum/P. boydii* (6 isolates), an XTT-based colorimetric microbroth assay was used (45). The isolate-specific MIC and concentrations 4- and 16-fold higher than the determined MIC were used. The effect was compared to an untreated growth control. An inoculum of $2 \times 10^6$ CFU/mL was incubated together with the LLCZ drug concentrations in flat-bottomed 96-well plates at 35°C (final inoculum concentration, $1 \times 10^6$). The growth was assessed after 0 h, 6 h, 12 h, 24 h, and 48 h of incubation with the drug. Two hours before the end of incubation, 50 $\mu$L of 2.5 mg/mL XTT (Santa Cruz Biotechnology, Dallas, TX, USA) plus 125 $\mu$M menadione (Sigma) solution was added to each well. Aliquots (150 $\mu$L) of the suspensions were transferred from each well onto new 96-well plates with a U-shaped bottom. The $OD_{492}$ was measured using a microplate reader (Sunrise, Tecan, Switzerland).

**(iii) Biofilm assays.** The biofilm analysis methods were previously described (23, 32, 34). In brief, a final inoculum concentration of $1 \times 10^6$ CFU/mL was used for all biofilm assays. LLCZ was used at the previously determined isolate-specific MICs, as well as at sub-MICs: $0.5\times$, $0.25\times$, and $0.125\times$ the specific MIC value. An untreated growth control was also included. Aliquots (1 mL) of the suspensions were incubated in 24-well plates at 35°C. LLCZ was added at different time points after the initial incubation: 0 h ($t_0$), 2 h ($t_2$) and 48 h ($t_{48}$). The $t_0$ group was directly incubated with LLCZ for 48 h, while for the $t_2$ and $t_{48}$ groups, the biofilms were washed with $1\times$ PBS and incubated with LLCZ after initial incubation times of 2 h and 48 h, respectively. Afterward, the biofilms were washed with $1\times$ PBS and analyzed via XTT and crystal violet (CV) assays. For the XTT assay, 500 $\mu$L of a solution of XTT (0.5 mg/mL) plus 25 $\mu$M menadione was pipetted into each well of the plates, which were then incubated at 36°C for 3 h in the dark. For the crystal violet assay, the biofilms were stained with a 0.01% CV solution for 20 min, air-dried overnight at room temperature, and then incubated for 30 min in a 30% acetic acid solution. Aliquots (100 $\mu$L) from each well were transferred into a U-shaped 96-well plate, and the optical density was measured in a plate reader at 492 nm (XTT assay) and 620 nm (CV).

**(iv) Confocal laser scanning microscopy.** The effect of LLCZ against biofilm formation was visualized via confocal laser scanning microscopy (CLSM) using the Elyra LSM 710 instrument (Zeiss, Oberkochen, Germany) with a laser at 405 nm and a 20-fold magnification objective. Suspensions were incubated in $\mu$-Slide 8-well glass-bottom plates (ibidi GmbH, Gräfelfing, Germany), as described above. The biofilms were fixed with 100% methanol for 1 min. Images were processed using ZEN black software (Zeiss).

***In vivo* assays. (i) *Galleria mellonella* larva handling.** *Galleria mellonella* (larval stage) was used as the *in vivo* model. The standardized protocol for handling of the larvae was previously published (39). Briefly, the larvae were weighed and separated into groups. The groups were injected in the prolegs with a corresponding fungal inoculum or treatment in volumes of 10 $\mu$L (for 300 $\pm$ 50 mg larvae), 13.33 $\mu$L (for 400 $\pm$ 50 mg larvae), and 16.66 $\mu$L (for 500 $\pm$ 50 mg larvae) with the help of a syringe pump (SyringePumpPro model LA-100; Landgraph Laborsysteme HLL GmbH, Langenhagen, Germany). The infected larvae were subsequently incubated at 37°C over 10 days with daily survival monitoring.

**(ii) Infection treatment.** The treatment of the larvae infected with *L. prolificans*, *S. apiospermum*, and *P. boydii* with a single dose of LLCZ was tested. Based on a previous infection assay (see Fig. S1 in the supplemental material), the caterpillars were injected in the prolegs with an inoculum of $2 \times 10^7$ CFU/mL from each pathogenic organism. LLCZ was injected at a concentration of 5 mg/kg body weight, equaling 152 mg/L (36). In parallel, other groups of larvae were treated with VCZ, which served as a control. The concentration used for VCZ was 160 mg/L, which corresponds to a recommended therapeutic dose of 6 mg/kg in humans (37). The larvae were injected with the fungal inoculum, incubated for 48 h, and then treated with the antifungals. At treatment, a different proleg was chosen for injection, to avoid any unnecessary stress of the larvae. Four different control groups were used: a noninjection control; an injection control, where larvae were injected 2 times with PBS; an infection control, where infected larvae were injected with PBS; and a treatment control, where the infected larvae were treated with VCZ.

**(iii) Recultivation of the organisms.** Recultivation of the pathogens from dead larvae was performed as previously described (39). The dead larvae were disintegrated using the MagNA lyser (Roche, Grenzach-Wyhlen, Germany) and consecutively plated on Sabouraud agar to confirm the reisolation of the pathogen.

**Statistical analysis. (i) *In vitro*.** All experiments were performed in triplicate. The growth percentage of each treated well was calculated in comparison to that of the untreated control. A statistical analysis of the growth percentage was carried out using the program GraphPad Prism 9 (GraphPad Software Inc., La Jolla, CA, USA). The statistical significance level was determined using Dunnett's multiple-comparison test; significance was determined at $P < 0.05$ (*, $P < 0.05$; **, $P < 0.01$; ***, $P < 0.001$; and ****, $P < 0.0001$).

**(ii) *In vivo*.** All experiments were performed in triplicate. The survival data were analyzed using the program GraphPad Prism 9 (GraphPad Software Inc.). Survival curves were plotted using the Kaplan-Meier estimator. To compile the results from multiple repetitions, the raw data were calculated as a

single experiment on the datasheet. The statistical significance of the survival curves was determined with a log-rank (Mantel-Cox) test; significance was set at $P < 0.05$ (*, $P < 0.05$; **, $P < 0.01$).

## SUPPLEMENTAL MATERIAL

Supplemental material is available online only.

**SUPPLEMENTAL FILE 1**, PDF file, 0.8 MB.

## ACKNOWLEDGMENTS

We hereby thank the Medicines for Malaria Venture (MMV) for their generous donation of the Pandemic Response Box; the Imaging Center Essen (IMCES), Faculty of Medicine, University Hospital Essen, University of Duisburg-Essen, for assistance and the use of their imaging facilities; and David Killengray, a native English speaker, for proofreading the manuscript. We further acknowledge support by the Open Access Publication Fund of the University of Duisburg-Essen.

This work was supported by internal funding. We declare no conflicts of interest.

Conceptualization, L.K. and D.-T.F.; methodology, D.-T.F. and L.K.; validation, D.-T.F., L.K., S.D., and U.S.; formal analysis, D.-T.F.; resources, P.-M.R. and J.F.M.; data curation, D.-T.F.; writing – original draft preparation, D.-T.F.; writing – review and editing, L.K., P.-M.R., J.S., S.D., U.S., and J.F.M.; visualization, D.-T.F.; supervision, L.K., P.-M.R., and J.S.; project administration, L.K. and P.-M.R. All authors have read and agreed to the published version of the manuscript.

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
