## [Reviewer comments · Microbiology Spectrum]

Microbiology Spectrum

***In vitro* and *in vivo* effect of the imidazole luliconazole against *Lomentospora prolificans* and *Scedosporium* spp.**

Dan-Tiberiu Furnica, Silke Dittmer, Ulrike Scharmann, Jacques Meis, Jörg Steinmann, Peter-Michael Rath, and Lisa Kirchhoff

Corresponding Author(s): Dan-Tiberiu Furnica, Universitätsklinikum Essen Institut für Medizinische Mikrobiologie

Review Timeline:

Submission Date:	December 20, 2022
Editorial Decision:	February 13, 2023
Revision Received:	March 4, 2023
Accepted:	March 14, 2023

Editor: Dimitrios Kontoyiannis

Reviewer(s): The reviewers have opted to remain anonymous.

Transaction Report:

DOI: <https://doi.org/10.1128/spectrum.05130-22>

February 13, 2023

Dr. Dan-Tiberiu Furnica
Universitätsklinikum Essen Institut für Medizinische Mikrobiologie
Medical Microbiology
Virchowstraße 179
Essen, Nordrhein-Westfalen 45147
Germany

Re: Spectrum05130-22 (*In vitro* and *in vivo* effect of the imidazole luliconazole against *Lomentospora prolificans* and *Scedosporium* spp.)

Dear Dr. Dan-Tiberiu Furnica:

Link Not Available

Sincerely,

Dimitrios Kontoyiannis

Journals Department
Reviewer comments:

Reviewer #1 (Comments for the Author):

The manuscript (Spectrum05130-22) entitled "In vitro and in vivo effect of the imidazole luliconazole against *Lomentospora prolificans* and *Scedosporium* spp" investigated the in vitro activity of the imidazole luliconazole (LLCZ) against *Scedosporium apiospermum* (including its teleomorph *Pseudallescheria boydii*) and *Lomentospora prolificans* and assessed its efficacy in a *G. mellonella* model.

The manuscript is very well-conducted and potentially interesting for publication, however, there are some aspects, which must be addressed before consideration.

The manuscript requires revisions with gross English editing preferably by a native speaker/editor

I have no major comments but I would like to suggest you highlight the following questions.

Authors should at least speculate what could be the importance of their findings in practical application and give some future perspectives for future studies.

-My only minor issue to consider is to maybe give some indication of why this compound was targeted as there is no context to its use in these circumstances.

The pharmacologic target and mode of action are unknown. For a better explanation

• Khanna D, Bharti S. 2014. Luliconazole for the treatment of fungal infections: an evidence-based review. *Core Evid* 9:113-124. following reference

-Luliconazole does not use for the treatment of invasive fungal infections. However, Luliconazole is a newly FDA-approved topical imidazole for the treatment of superficial mycoses such as tinea pedis, tinea cruris, tinea corporis, and onychomycosis, It also has broad in vitro activity against other non-dermatophyte fungal pathogens such as *Candida* spp., *Cryptococcus neoformans*, *Malassezia* spp, *Fusarium* spp. and *Aspergillus* spp

So please explain costs, side effects, lifetime, etc

The following references would be helpful. Please consider this publication for your manuscript

• Baghi N, Shokohi T, Badali H, et al. In vitro activity of new azoles luliconazole and lanoconazole compared with ten other antifungal drugs against clinical dermatophyte isolates. *Med Mycol*. 2016;54(7):757-763.

• Rezaei-Matehkolaei A, Khodavaisy S, Alshahni MM, et al. Antifungal activity of novel triazole efinaconazole and five comparators against dermatophyte isolates. *Antimicrob Agents Chemother*. 2018;62(5):e02423

• Nyuykonge B, Lim W, van Amelsvoort L, Bonifaz A, Fahal A, Badali H, Abastabar M, Verbon A, van de Sande W. Eumycetoma causative agents are inhibited in vitro by luliconazole, lanoconazole and ravuconazole. *Mycoses*. 2022 Jun;65(6):650-655. doi: 10.1111/myc.13442. Epub 2022 May 6. PMID: 35398930; PMCID: PMC9321754.

Why you did not use other antifungal drugs as comparator drugs in this experiment?

S. apiospermum/*P. boydii* strains were low? Why you did not include more diverse isolates from different sources? However, the data is interesting. This should at least be presented as a limitation of the study.

What was the clinical breakpoint?

Line169 2.5.1 *G. mellonella* larvae handling

Was Luliconazole tolerated in *G. mellonella* larvae"? if yes, it would be useful to use a murine model as well.

change the "imidazole luliconazole" to "luliconazole" in the entire manuscript

telomorph?? correct it

Line 238-240: does not make sense, please rephrase it

Do you advise to do always AFST when isolating *Lomentospora prolificans* and *Scedosporium* spp from an invasive infection?

Line 115: dimethyl sulfoxide (DMSO): %??

Why didn't include more genetically diverse or distant isolates? This should at least be presented as a limitation of the study.

Although your collection is likely genetically similar,

What about GM MIC??, Mode of MIC

Please update references

Reviewer #2 (Comments for the Author):

The study investigated the in vitro and in vivo activity of the imidazole luliconazole (LLCZ) against a limited set of 37 isolates of *Scedosporium apiospermum* and *Lomentospora prolificans*. The authors used XTT based colorimetric microbroth assay to assess the effect of LLCZ against the planktonic growth of *L. prolificans* for 8 isolates, and for 6 isolates of *S. apiospermum*/*P. boydii*. Further they used *Galleria mellonella* infection model for in vivo treatment assays. The paper is well written and reports novel findings of the anti-biofilm effect of LLCZ in *Scedosporium* spp. The methodology used is sound. A minor comments listed below needs clarification:

1. How the isolates were selected for the Growth kinetics assay- based on MIC data or patient background data? As several isolates are from cystic fibrosis patient populations which may represent colonization. A brief clinical detail needs to be included.
2. For growth kinetics study inoculum of 2×10^6 CFU/mL was used which seems to be on the high side-how was the inoculum size calculated?
3. LCZ is so far available for topical application, therefore limited studies have tested its activity in animal models. This study gives support that if LCZ is made available in oral formulation it may be a potent drug for azole resistant fungi. The authors may consider emphasising this aspect in discussion.
4. In vivo, a single dose of LLCZ increased the survival rate of the larvae by 40 % and 20 % for *L. prolificans* and *Scedosporium* spp which is still not very promising. This reviewer feels that the reproducibility is a problem in the *Galleria* model and that aspect needs to be discussed.
5. Line 107 please correct H2O

6. Line 241-242 needs modification.

Staff Comments:

Preparing Revision Guidelines

Please return the manuscript within 60 days; if you cannot complete the modification within this time period, please contact me. If you do not wish to modify the manuscript and prefer to submit it to another journal, please notify me of your decision immediately so that the manuscript may be formally withdrawn from consideration by Microbiology Spectrum.

**Response to the reviewers of the manuscript:
,In vitro and in vivo effect of the imidazole luliconazole against
Lomentospora prolificans and Scedosporium spp.'**

We would hereby like to thank the editor for considering our manuscript. Please find attached a point-to-point reply on the reviewer's comments. All changes were marked up in the revised version of the manuscript.

Reviewer #1:

The manuscript requires revisions with gross English editing preferably by a native speaker/editor

The manuscript was sent for professional English editing by a native speaker.

I have no major comments but I would like to suggest you highlight the following questions. Authors should at least speculate what could be the importance of their findings in practical application and give some future perspectives for future studies.

We agree with the reviewer that the importance of our findings could be better underlined. That is why a paragraph regarding the practical applications of LLCZ/ perspectives for future studies was added to the revised manuscript (lines 316-322).

-My only minor issue to consider is to maybe give some indication of why this compound was targeted as there is no context to its use in these circumstances.

We acknowledge the fact that little context was given for the use of this drug in this study. Now we have added a paragraph in the introduction which explains why LLCZ was chosen in this context (lines 55-60).

The pharmacologic target and mode of action are unknown. For a better explanation

• Khanna D, Bharti S. 2014. Luliconazole for the treatment of fungal infections: an evidence-based review. Core Evid 9:113-124. following reference

-Luliconazole does not use for the treatment of invasive fungal infections. However, Luliconazole is a newly FDA-approved topical imidazole for the treatment of superficial mycoses such as tinea pedis, tinea cruris, tinea corporis, and onychomycosis. It also has broad *in vitro* activity against other non-dermatophyte fungal pathogens such as *Candida* spp., *Cryptococcus neoformans*, *Malassezia* spp, *Fusarium* spp. and *Aspergillus* spp

We agree with the reviewer on the fact that LLCZ has no current clinical application for IV use and that the antifungal effect of this drug been tested *in vitro* and in non-clinical contexts. We believe that it should be made very clear that this drug, although FDA approved, has not been released for systemic use and that its mode of action is unknown (lines 60-63). We now included the mentioned reference in the discussion.

So please explain costs, side effects, lifetime, etc

We now included a more detailed introduction and discussion part regarding the drug luliconazole in the revised manuscript (lines 67-79 and 308-322).

The following references would be helpful. Please consider this publication for your manuscript

- Baghi N, Shokohi T, Badali H, et al. In vitro activity of new azoles luliconazole and Ianoconazole compared with ten other antifungal drugs against clinical dermatophyte isolates. *Med Mycol.* 2016;54(7):757-763.
- Rezaei-Matehkolaei A, Khodavaisy S, Alshahni MM, et al. Antifungal activity of novel triazole efinaconazole and five comparators against dermatophyte isolates. *Antimicrob Agents Chemother.* 2018;62(5):e02423
- Nyuykonge B, Lim W, van Amelsvoort L, Bonifaz A, Fahal A, Badali H, Abastabar M, Verbon A, van de Sande W. Eumycetoma causative agents are inhibited in vitro by luliconazole, Ianoconazole and ravuconazole. *Mycoses.* 2022 Jun;65(6):650-655. doi: 10.1111/myc.13442. Epub 2022 May 6. PMID: 35398930; PMCID: PMC9321754.

We would like to thank the reviewer for the suggested references. The first two references present the *in vitro* effect of the drug against dermatophytes, while the third also presents some *in vivo* data with the *G. mellonella* model and touches upon the drawbacks of LLCZ. We have included them in the revised manuscript.

Why you did not use other antifungal drugs as comparator drugs in this experiment?

The effect of several antifungal drugs such as voriconazole, amphotericin B, micafungin and olorofim on *Lomentospora prolificans* has been previously shown by our working group for the isolates used in this manuscript.

Kirchhoff L, Dittmer S, Weisner A-K, Buer J, Rath P-M, Steinmann J. 2020. Antibiofilm activity of antifungal drugs, including the novel drug olorofim, against *Lomentospora prolificans*. *Journal of Antimicrobial Chemotherapy* 75:2133-2140.

We considered that it would be redundant to repeat the experiments, since the same methods and isolates were used in previous publications showing the effect of different antifungal drugs. We revised to this data.

S. apiospermum/ P. boydii strains were low? Why you did not include more diverse isolates from different sources? However, the data is interesting. This should at least be presented as a limitation of the study.

Due to the lack of availability of rare moulds such as *S. apiospermum/ P. boydii* the strain collection of our laboratory is not very large. In the original manuscript, this was presented as a limitation of the study. In the revised manuscript, we made sure to point this out more clearly (lines 329-334).

What was the clinical breakpoint?

The clinical breakpoints for luliconazole are not yet defined.

Line169 2.5.1 *G. mellonella* larvae handling

Was Luliconazole tolerated in *G. mellonella* larvae"? if yes, it would be useful to use a murine model as well.

Luliconazole was tolerated well by *G. mellonella*. A reference to a previous paper where this is demonstrated has been made in the manuscript. We agree with the reviewer that a murine model would be more advantageous, especially because only a few publications have used luliconazole in mice and rats. A recommendation for a perspective future use of a murine model has been added in the discussion (lines 328-329 and 334).

change the "imidazole luliconazole" to "luliconazole" in the entire manuscript

We now changed imidazole luliconazole to luliconazole.

telomorph?? correct it

We thank the reviewer for this correction and edited the spelling mistake.

Line 238-240: does not make sense, please rephrase it

Lines have been rephrased.

Do you advise to do always AFST when isolating *Lomentospora prolificans* and *Scedosporium* spp from an invasive infection?

From a clinical point of view an AFST for *Lomentospora* and *Scedosporium* from a sterile patient sample should be performed in a lab with high expertise or reference laboratory. This is also stated in the EQUAL score for *Scedosporium/Lomentospora* (<https://doi.org/10.1093/jac/dkab355>).

Line 115: dimethyl sulfoxide (DMSO): %??

The final percentage of DMSO after suspension of LLCZ in RPMI is 1%. We updated the methods section (lines 103-104)

Why didn't include more genetically diverse or distant isolates? This should at least be presented as a limitation of the study. Although your collection is likely genetically similar,

As previously stated, our strain collection encompasses a limited number of strains but we don't agree with the assumption that they are likely genetically similar because all were from different patients from different places in the Netherlands collected over several sequential years.

What about GM MIC??, Mode of MIC

In the paper we have presented only a calculated MIC₉₀ of 0.25 µg/ml. As the MIC₅₀ was 0.06 µg/ml, the calculated geometric mean MIC was 0.1224 µg/ml. We made sure to include this in the results section. The MICs of the isolates were determined via broth microdilution and visual reading of the wells after the method used by EUCAST. We updated the methods section to make this clearer (lines 113-114 and 209-211).

Please update references

References were updated

Reviewer #2:

1. How the isolates were selected for the Growth kinetics assay- based on MIC data or patient background data? As several isolates are from cystic fibrosis patient populations which may represent colonization. A brief clinical detail needs to be included.

For the experiments, one reference strain for each separate organism (*L. prolificans*: F125; *S. apiospermium*/*S. boydii*: M222/M356) were selected. There were no specific selection criteria

for the isolates, the process of selection for the clinical isolates being random. Clinical data regarding these isolates, except that they were from CF patients has not been made available to us. We included this information in the revised manuscript (line 88). We agree with the referee that they were most probably representing colonization since invasive infections in CF patients is a rare event.

2. For growth kinetics study inoculum of 2×10^6 CFU/mL was used which seems to be on the high side-how was the inoculum size calculated?

In the growth kinetics study, a final inoculum of 1×10^6 CFU/mL was used due to a final 1:2 dilution when adding the drug. The inoculum of 1×10^6 CFU/ml is used in our work group since it accelerates the planktonic growth and biofilm formation of *L. prolificans* and *Scedosporium* spp., allowing the organisms to grow in periods of up to 48 h.

3. LCZ is so far available for topical application, therefore limited studies have tested its activity in animal models. This study gives support that if LCZ is made available in oral formulation it may be a potent drug for azole resistant fungi. The authors may consider emphasising this aspect in discussion.

We would like to thank the reviewer for their observation. While the FDA does not yet approve an oral formulation of this drug, we believe that our findings could increase the support for developing a viable formulation of LLCZ, which works against azole resistant fungi. A paragraph regarding potential future uses has been added in the discussion sections of the revised manuscript (lines 308-322).

4. *In vivo*, a single dose of LLCZ increased the survival rate of the larvae by 40 % and 20 % for *L. prolificans* and *Scedosporium* spp which is still not very promising. This reviewer feels that the reproducibility is a problem in the Galleria model and that aspect needs to be discussed.

We acknowledge the fact that despite its benefits, the *G. mellonella* has several drawbacks, including its dependency on the quality of the larvae which may cause certain deviations from the original results upon reproduction of the experiment. We discussed this issue in the revised manuscript (lines 325-329)

5. Line 107 please correct H2O

H2O was modified to H₂O

6. Line 241-242 needs modification.

The line has been modified

March 14, 2023

Dr. Dan-Tiberiu Furnica
Universitätsklinikum Essen Institut für Medizinische Mikrobiologie
Medical Microbiology
Virchowstraße 179
Essen, Nordrhein-Westfalen 45147
Germany

Re: Spectrum05130-22R1 (*In vitro* and *in vivo* effect of the imidazole luliconazole against *Lomentospora prolificans* and *Scedosporium* spp.)

Dear Dr. Dan-Tiberiu Furnica:

Your manuscript has been accepted, and I am forwarding it to the ASM Journals Department for publication. You will be notified when your proofs are ready to be viewed.

Sincerely,

Dimitrios Kontoyiannis
Editor, Microbiology Spectrum
